# Slower Growth during Lactation Rescues Early Cardiovascular and Adipose Tissue Hypertrophy Induced by Fetal Undernutrition in Rats

**DOI:** 10.3390/biomedicines10102504

**Published:** 2022-10-07

**Authors:** Pilar Rodríguez-Rodríguez, Ignacio Monedero-Cobeta, David Ramiro-Cortijo, Sophida Puthong, Begoña Quintana-Villamandos, Alicia Gil-Ramírez, Silvia Cañas, Santiago Ruvira, Silvia M. Arribas

**Affiliations:** 1Department of Physiology, Faculty of Medicine, Universidad Autónoma de Madrid, 28029 Madrid, Spain; 2Food, Oxidative Stress and Cardiovascular Health (FOSCH) Research Group, Universidad Autónoma de Madrid, Ciudad Universitaria de Cantoblanco, 28049 Madrid, Spain; 3Department of Physiology, Khon Kaen University, Khon Kaen 40002, Thailand; 4Department of Anesthesiology, Hospital General Universitario Gregorio Marañón, 28007 Madrid, Spain; 5Department of Agricultural Chemistry and Food Science, Faculty of Science, Universidad Autónoma de Madrid, Ciudad Universitaria de Cantoblanco, 28049 Madrid, Spain

**Keywords:** fetal undernutrition programming, cross-fostering, lactation period, cardiovascular hypertrophy, adipose tissue browning

## Abstract

Low birth weight (LBW) and accelerated growth during lactation are associated with cardiometabolic disease development. LBW offspring from rats exposed to undernutrition during gestation (MUN) develops hypertension. In this rat model, we tested if slower postnatal growth improves early cardiometabolic alterations. MUN dams were fed *ad libitum* during gestation days 1–10, with 50% of the daily intake during days 11–21 and *ad libitum* during lactation. Control dams were always fed *ad libitum.* Pups were maintained with their own mother or cross-fostered. Body weight and length were recorded weekly, and breastmilk was obtained. At weaning, the heart was evaluated by echocardiography, and aorta structure and adipocytes in white perivascular fat were studied by confocal microscopy (size, % beige-adipocytes by Mitotracker staining). Breastmilk protein and fat content were not significantly different between groups. Compared to controls, MUN males significantly accelerated body weight gain during the exclusive lactation period (days 1–14) while females accelerated during the last week; length growth was slower in MUN rats from both sexes. By weaning, MUN males, but not females, showed reduced diastolic function and hypertrophy in the heart, aorta, and adipocytes; the percentage of beige-type adipocytes was smaller in MUN males and females. Fostering MUN offspring on control dams significantly reduced weight gain rate, cardiovascular, and fat hypertrophy, increasing beige-adipocyte proportion. Control offspring nursed by MUN mothers reduced body growth gain, without cardiovascular modifications. In conclusion, slower growth during lactation can rescue early cardiovascular alterations induced by fetal undernutrition. Exclusive lactation was a key period, despite no modifications in breastmilk macronutrients, suggesting the role of bioactive components. Our data support that lactation is a key period to counteract cardiometabolic disease programming in LBW and a potential intervention window for the mother.

## 1. Introduction

Adverse environmental conditions during the intrauterine period increase the risk to develop cardiometabolic diseases (CMD), a process known as fetal programming [1,2]. Exposure to the fetus to insufficient nutrients is one of the key programming factors, widely demonstrated in human populations by the association between low birth weight (LBW) and higher rates of hypertension, coronary heart disease, or type 2 diabetes mellitus [3,4,5]. Fetal programming has also been validated in animal models [6,7,8]. The postnatal period seems to be a second critical window for programming, particularly after exposure to a fetal stressor. During this period, LBW individuals frequently experience a rapid weight gain, and compelling evidence indicates that this growth pattern is also deleterious. Adolescents born prematurely with a quick postnatal growth have a lower flow-mediated-endothelial dilatation compared to those with slower weight gain and a higher risk of hypertension [8]; quick growth and elevated central adiposity in infancy also contribute to the future development of metabolic disorders [9,10]. A systematic review highlights that rapid postnatal catch-up growth is a more important factor than LBW alone for the development of cardiovascular disease and its risk factors [11].

Nutrition in LBW individuals represents a challenge to ensure a postnatal growth pattern adequate to achieve brain and bone development while preventing CMD [12]. In this context, the lactation period has been proposed as a critical window to counteract alterations initiated during fetal life [13], since human breast milk is recognized as the gold standard for infant nutrition tailored to provide the nutrients and bioactive factors for healthy growth [14]. Breastfeeding provides protection against CMD, is associated with lower blood pressure levels and a lower risk of overweight in children [15,16], and a systematic review evidences slower rates of weight gain in preterm and LBW infants fed with breastmilk compared to the formula [17].

Animal models provide the tools to analyze the influence of both intrauterine and lactation periods in a controlled environment. In a rat model of fetal programming induced by maternal undernutrition during gestation (MUN), we have demonstrated that LBW offspring accelerate growth during lactation, reaching the same body weight as control counterparts by wearing. This catch-up trajectory in MUN rats leads to excessive growth of cardiovascular organs in males [18,19] and white adipose tissue (WAT) deposits in both sexes [20].

In this study, we aim to demonstrate in an animal model of LBW if a slower growth during the lactation period can rescue the cardiometabolic alterations induced by undernutrition. To test this hypothesis, in MUN rats we have modified the lactation environment using a cross-fostering protocol and analyzed the impact on (1) body weight and length growth pattern during the first three weeks of postnatal life, (2) milk macronutrient composition, and (3) heart, aorta and perivascular adipose tissue growth and cardiac function at weaning, assessing the influence of sex in the abovementioned parameters.

## 2. Materials and Methods

Experiments were performed on Sprague Dawley rats from the colony bred at the Animal House of Universidad Autónoma de Madrid (ES-28079-0000097). This study was performed in accordance with European Union Guidelines for the Care and Use of Laboratory Animals (Directive 2010/63/EU and Spanish RD 1201/2005) and was approved by the Ethics Review Board of Universidad Autónoma de Madrid and by the local government (Comunidad Autónoma de Madrid, Spain; PROEX 004-19, approved on 7 May 2019).

The rats were housed in buckets 36.5 × I21.5 × 18.5 cm (length×width×height) on aspen wood bedding, maintained under controlled conditions of temperature and 12/12 light/dark photoperiod, and the welfare of the animals was monitored daily by staff. The animals were fed with a EuroRodent breeding diet (5LF5; Labdiet, Madrid, Spain) containing 22.0% protein, 4.4% fat, 55.0% carbohydrates, 4.1% fiber, and 5.4% mineral (sodium content 0.26%) and drinking water were provided *ad libitum*.

### 2.1. Maternal Undernutrition (MUN) Model and Cross-Fostering Protocol

We used a rat model of fetal programming induced by maternal global nutrient restriction during the second part of gestation, as previously described [21]. After observation of sperm in the vaginal smear (gestation = day 0) the dam was allocated to either MUN or the control group. Control dams were fed *ad libitum* throughout pregnancy and lactation; MUN dams were fed *ad libitum* from day 1–10 and with 50% of the averaged control daily intake (previously calculated as 24 g/day) from day 11 to the end of gestation, returning to *ad libitum* diet during lactation. 24 h after birth the pups were sexed and weighed individually and the litter was standardized to 12 individuals, 6 males, and 6 females if possible (smaller litters were not used for this work, being used for other studies). Control and MUN litters were left with their mothers or cross-fostered.

**Cross-fostering protocol**. We used a protocol similar to [22] with some modifications. Two dams were mated at the same time, ensuring the same gestation day 0, and those with birth on the same day were cross-fostered. On postnatal day 1, the entire litter was exchanged to the foster’s dam cage with bedding from the original mother to avoid rejection. The litter was standardized to 12 individuals and the pups were sexed. The rats were not individually marked to help acceptance (no rejection was detected in any of the groups).

The name of the groups was “pup type–on–mother type” (i.e., M-on-C refers to a pup from a MUN dam fostered by a control mother). Four experimental groups were studied: control-on-control (C-on-C; n = 6 mothers), MUN-on-MUN (M-on-M; n = 6 mothers); control-on-MUN (C-on-M, n = 4 mothers), MUN-on-control (M-on-C; n = 4 mothers).

### 2.2. Experimental Protocols

The entire offspring from the four groups were weighted and head-to-tail length measured at days 1, 14, and 21 of the lactation periods with an analytical balance (Conecta, Barcelona, Spain) and a digital caliper (Conecta, Barcelona, Spain), respectively. The body weight gain from day 1 to 14 was calculated as Formula (1). A similar formula was used to assess body weight gain during the last week and length gain in both lactation periods. Weigh gain was expressed as g/day and length gain as cm/day.
(1)$$Weight\;gain=\frac{\left(Weight_{day\;14}-Weight_{day\;1}\right)}{number\;of\;days}\;$$

Breast milk was collected from MUN and control dams on days 12 and 14 of lactation (see protocol below). On day 21 (weaning), 1–2 males and 1–2 females from each litter were studied and the remaining animals were used for other studies according to the 3Rs rule for experimental animals. Firstly, cardiac structure and function were evaluated by echocardiography. Thereafter, the rats were placed in a CO_2_ chamber and killed by exsanguination. Then, the thoracic aorta and mesenteric bed were removed and stored in a cold saline solution. The aorta was cleaned, and the aortic arch and immediate region were fixed in 4% paraformaldehyde solution (PFA). Perivascular adipocytes from the mesenteric bed were also dissected and fixed in PFA. Both tissues were stored for later confocal microscopy analysis. Figure 1 shows a summary of experimental protocols.

### 2.3. Breast Milk Collection and Measurements

Breast milk was collected on day 12th and day 14th of lactation (the period of maximal milk yield) and was pooled, allowing the mother and pups to recover during day 13. On the day of collection, the pups were separated from the mother for 3 h; the dam was anesthetized with isoflurane inhalation anesthesia (1000 mg/g; Isoflutek, Karizoo Lab., Barcelona, Spain) and injected with 0.1–0.15 mL of oxytocin i.p. depending on rat body weight (10 UI/mL; Syntocinon, Mylan Products Ltd., Hertfordshire, UK). A 1.0–1.5 mL volume of breast milk was obtained by gentle massage of the mammary glands and collection by a glass micropipette in Eppendorf tubes. The dam was milked for a maximum of 1 h. Each breastmilk sample was divided into two aliquots, one to assess total fat and the second one to assess total proteins.

**Total fat content.** Fat was analyzed by the Mojonnier method [23] with modifications. Briefly, a 0.5 mL volume of breast milk was mixed with 0.5 mL of ammonium hydroxide, followed by the addition of 50 µL of ethanolic phenolphthalein solution (0.5% *w*/*v*) and shaken. Thereafter, 2.5 mL of ethanol, 2.5 mL of petroleum ether, and 2.5 mL of ethyl ether were added and vigorously mixed for 30 s. Thereafter, the mix was centrifuged at 4000× *g* (3 min, RT) and the upper phase containing the fat was stored. This process was repeated three times with the remaining aqueous phase, adding the ethanol, ethyl ether, and petroleum ether. Thereafter, the fat from the 3 centrifugation was pooled, and to evaporate the ether solvent, it was placed overnight in a gravity convection oven at 50 °C, uncapped. Finally, the total fat content in the sample was measured by gravimetry and expressed as mg/mL of BM. A blank reaction was performed by substituting the 0.5 mL breast milk volume with H_2_O-Q.

**Total protein level.** Protein quantification was carried out in the defatted phase of breast milk using the Bradford method [24]. It has to be noted that centrifugated breast milk samples continue to contain proteins such as caseins. Briefly, 10 µL of BM (1:50 *v*/*v* in H_2_O-Q) were mixed in a microplate with a 200 µL volume of Coomassie-blue dye (1:4 *v*/*v* in H_2_O-Q; Bio-Rad Lab., Hercules, CA, USA). After shaking the mix for 1 min, the absorbance was measured at 595 nm in a microplate reader (Synergy HT Multimode; BioTek Instruments, Winooski, VT, USA). Total protein levels were expressed as mg of BSA/mL. BSA (Sigma-Aldrich, St. Louis, MO, USA) was used for the standard curve (range 0.1–0.5 mg/mL) and Coomassie-blue dye was substituted by H_2_O-Q for the blank curve.

### 2.4. Transthoracic Echocardiography

Transthoracic echocardiography was performed in 21 days-old rats using the VIVID q-system (GE Healthcare, Munich, Germany) equipped with a 13 MHz probe (12S-RS, GE Healthcare, Munich, Germany) as previously described [25]. Briefly, the rats were anesthetized by i.p injection with 80 mg/kg Ketamine hydrochloride (AuroMedics Pharma LLC., Dublin, Ireland) and 10 mg/kg Diazepam (Hospira, Inc., Lake Forest, IL, USA). The images were acquired with the animals in left lateral decubitus. M-mode imaging of the parasternal short-axis (papillary level) view allowed measurement of end-diastolic (mm) and end-systolic (mm) internal diameter, posterior wall thickness (mm), and interventricular septum thickness (mm) at diastole. Additionally, it was calculated the systolic functionality as the ejection fraction (%). The pulsed-wave Doppler early-to-late transmitral peak diastolic flow velocity ratio (E/A ratio; arbitrary units) was measured to assess diastolic function (E, mitral peak early-filling velocity and A, mitral peak flow velocity at atrial contraction). The transmitral flow velocity profile was determined by positioning a sample volume at the tip of the mitral valve on the para-apical long-axis view. The E-wave deceleration time was measured as the time interval between peak E-wave velocity and the point where the descending E-wave (or its extrapolation) intercepted the zero line. Values were determined by averaging the measurements of three consecutive cardiac cycles.

### 2.5. Aorta Structure by Confocal Microscopy

Aorta structure was assessed in a ring cut from the PFA fixed thoracic aorta proximal to the aortic arch. The ring was mounted intact on a slide provided with a small well, to avoid compression containing Citifluor AF2 mounting medium (Aname, Madrid, Spain), as previously described [19]. The ring was visualized with a ×20 objective with a laser scanning confocal microscope (Leica TCS SP2, Leica Microsystems, Barcelona, Spain) at Excitation = 488 nm/Emission = 500–560 nm, the wavelength at which elastin can be detected by its autofluorescence allowing to detect the medial layer. Single images were captured with a ×20 air objective, ensuring the elastic lamellae were clearly visible. Quantification was performed with FIJI software [26], measuring internal and external perimeters (µm); from these measurements medial cross-sectional areas (CSAs) were calculated in μm^2^, assuming the sections were circular.

### 2.6. Adipocyte Size and Browning

PFA-fixed adipocytes from the perivascular tissue of the mesenteric bed were incubated with Mitotracker red FM (Invitrogen, ThermoFisher, Madrid, Spain) prepared in phosphate buffer saline (PBS) to stain mitochondria-positive cells [27], evaluating the percentage of beige adipocytes, since they are rich in mitochondria. Adipocytes were incubated for 60 min (1:4000 *v*/*v*, RT, darkness) and washed 3 times for 10 min in PBS. Thereafter, they were incubated for 15 min in a solution of DAPI (Invitrogen, ThermoFisher, Madrid, Spain) to stain nuclei (1:500 *v*/*v* from stock solution, RT, darkness,) followed by 3 washes in PBS for 10 min each at RT. The samples were mounted intact on a slide with a small well and were visualized with a Laser Scanning Confocal Microscope (Leica TCS SP2, Leica Microsystems, Barcelona, Spain). Five regions were randomly chosen based on DAPI wavelength and single images were captured with a ×40 objective at three wavelengths: Excitation = 405 nm/Emission = 410–475 nm to visualize nuclei, Excitation = 488 nm/Emission = 500–560 nm to visualize adipocyte size by autofluorescence [20], and Excitation = 581 nm/Emission = 644 nm to visualize mitochondria-positive cells.

The quantification of adipocyte size and proportion of mitochondria-positive cells was performed by FIJI software. The total number and area occupied by the adipocytes were quantified in the Excitation = 488 nm/Emission = 500–560 nm wavelength images (autofluorescence), and the average adipocyte size was calculated in each image. To assess Mitotracker-positive cells in perivascular WAT, the area occupied by adipocytes was measured in autofluorescent images, Mitotracker-positive areas were measured in the same region in the red wavelength images, and the percentage was calculated.

### 2.7. Statistical Analysis

Statistical analysis was performed with R software (version 3.6.0, 2018, R Core Team, Vienna; Austria) within the R Studio interface using *rio, dplyr, compareGroups, ggpubr, devtools,* and *ggplot2* packages. Data was expressed as the median and interquartile range [Q1; Q3]. The differences in macronutrients of breast milk between control and MUN were performed by Mann-Whitney’s U test. The Kruskal–Wallis test by ranks was used to test the independent experimental groups. The significant Kruskal–Wallis test was used with Dunn’s test with directed pairwise comparison using C-on-C as a reference group. Second, it was considered the comparison between M-on-M versus M-on-C to determine, if pups with prior fetal programming, the lactation could reverse the modification. A *p*-value (*p*) < 0.05 was considered significant.

## 3. Results

### 3.1. Proteins and Fats in Breast Milk

No differences between control and MUN were detected in breast milk proteins (Control = 125.0 [113.0; 129.0] mg eq. BSA/mL, n = 12 rats; MUN = 117.0 [111.0; 123.0] mg eq. BSA/mL, n = 10 rats; *p*-Value=0.539). Fat levels in MUN milk tended to be lower but did not reach statistical significance (Control = 98.5 [89.3; 113.0] mg/mL, n = 12 rats; MUN = 84.0 [66.7; 107.0] mg/mL, n = 10 rats; *p*-Value = 0.090).

### 3.2. Body Growth Gain

From day 1 to 14 of lactation (exclusive lactation period), male rats from the M-on-M group exhibited a significantly higher increase in body weight gain compared to the C-on-C group. MUN males fostered by a C mother (M-on-C group) did not show differences compared C-on-C group, being significantly smaller compared to M-on-M males. Males from the C-on-M group had a significantly lower body weight gain compared to C-on-C (Figure 2A). By contrast, during the first lactation period, M-on-M female rats did not show differences in body weight gain compared to C-on-C or compared to M-on-C. As observed in males, control females nourished by a MUN mother (C-on-M) showed a lower body weight gain compared to C-on-C females (Figure 2A).

From day 15 to 21, when the pups suckle from mothers and eat by themselves, male M-on-M did not show statistical differences in body weight gain compared to C-on-C, or M-on-C males. However, M-on-M females accelerated growth during this period, as shown by the higher weight gain compared to C-on-C female rats. M-on-C rats also had higher weight gain compared to C-on-C females (Figure 2B).

From day 1 to 14, males of M-on-M length gain tended to be smaller compared to the C-on-C group but did not reach statistical significance (*p*-Value = 0.083). Cross-fostered MUN males (M-on-C) had a significantly higher length gain compared to the C-on-C group. In female rats, no differences were detected in any of the experimental groups in length gain during the exclusive lactation period (Figure 2C).

From day 15 to 21 length gain was significantly lower in M-on-M males and females compared to C-on-C control sex-matched counterparts. M-on-C males maintained a lower length gain, while M-on-C females improved the length rate, not significantly different from C-on-C females (Figure 2D).

### 3.3. Perivascular Adipocyte Size and Type

WAT adipocyte size was assessed in perivascular fat from the mesenteric bed obtained from 21-day-old offspring. M-on-M males had significantly larger adipocytes compared to C-on-C rats (Figure 3A). Adipocytes from MUN males fostered by a control mother (M-on-C) were of similar size compared to C-on-C males and significantly smaller compared to M-on-M. Control males fostered by a MUN mother (C-on-M) also showed smaller adipocytes size compared to C-on-C males (Figure 3A).

Regarding females, M-on-M rats had similar adipocytes compared to C-on-C. M-on-C females showed significantly smaller adipocyte size compared to C-on-C females and also compared to M-on-M. No differences were detected between control rats fostered by a MUN mother (C-on-M) and C-on-C females (Figure 3A).

We used Mitotracker, a dye that stains mitochondria, to evaluate beige-type adipocytes within perivascular WAT. Mitotracker-positive cells showed an intense stain. M-on-M males had a smaller percentage of Mitotracker-positive cells compared to C-on-C controls (Figure 4). MUN males fostered by a control dam (M-on-C) had a similar % compared to C-on-C males, being larger compared to M-on-M offspring. Control rats fostered by a MUN mother had lower Mitotracker-positive stained areas compared to C-on-C males.

As observed in males, M-on-M females had significantly lower Mitotracker positive area compared to C-on-C females. MUN rats fostered by a control mother (M-on-C) had a similar % compared to C-on-C offspring, being significantly larger compared to M-on-M rats (Figure 4).

### 3.4. Heart Structure and Function

Heart structure and function were assessed by transthoracic echocardiography at the end of the lactation period. In male M-on-M rats, posterior wall thickness tended to be higher and interventricular septum was significantly higher compared to C-on-C males. MUN males fostered by a control mother (M-on-C) did not show differences with C-on-C males. Female pups did not show significant differences in any of the studied parameters (Figure 5a,b).

Systolic function, assessed as ejection fraction, was not different between groups (Figure 5C). However, diastolic function, assessed as the ratio of E/A waves, was significantly reduced in M-on-M males compared to C-on-C counterparts. No significant differences were found in MUN males fostered by a C mother (M-on-C) compared to C-on-C (Figure 5D). Female M-on-M did not show significant differences in any of these parameters compared to C-on-C. However, the E/A ratio was significantly decreased in female M-on-C compared to female M-on-M (Figure 5D).

End-systolic and end-diastolic diameters were not significantly different between C-on-C and any of the other groups (Table 1).

### 3.5. Thoracic Aorta Structure

In M-on-M males’ internal and external diameter, and wall cross-sectional area (CSA) were significantly larger compared to C-on-C. In M-on-C males the studied parameters were not different from C-on-C males and external diameter and CSA were significantly smaller compared to M-on-M rats (Figure 6).

M-on-M females tended to have larger parameters compared to C-on-C, without statistical significance. However, MUN females fostered by a control mother (M-on-C) had a smaller external diameter and CSA than M-on-M rats (Figure 6).

## 4. Discussion

In this study, we aimed to evaluate the impact of lactation environment on body, cardiovascular and adipose tissue growth in rats with LBW induced by undernutrition during fetal life (MUN offspring). The main findings of the study are summarized in Table 2. During lactation, MUN offspring accelerate body weight gain, while length growth is slower. In males, this growth pattern occurs during the exclusive lactation period, while in females takes place during the last week before weaning, when the rats suckle and eat chow. By weaning, MUN males but not females, exhibit diastolic dysfunction, and heart, aorta, and perivascular white adipocyte hypertrophy; a lower proportion of beige-type adipocytes was found in MUN offspring from both sexes. These sex-dependent alterations may set the basis for the observed hypertension development in MUN males and can predispose to obesity in males and females. We also demonstrated that nursing MUN rats with a controlling mother rescues the hypertrophy alterations, in parallel with a slower weight gain, while nursing control pups, one MUN mother did not have a hypertrophic effect. These data suggest that both MUN rats, in the fetal and perinatal periods, are required to develop phenotypic alterations. In addition, the alterations programmed during the fetal period may be counteracted during lactation by reducing the growth rate. The period of exclusive lactation seems to be more relevant for the observed hypertrophy, even though no major modifications in breastmilk macronutrients were detected between MUN and Control dams, suggesting the role of breastmilk bioactive factors. In summary, the present data provide experimental evidence of the link between accelerated growth during early postnatal life in LBW individuals and higher-risk cardiometabolic diseases and support that modulation of growth during the lactation period can be an effective strategy to counteract alterations induced during fetal life.

Catch-up growth has been proposed to be deleterious for cardiometabolic health in individuals with LBW. We addressed the role of growth during lactation in a rat model of LBW induced by fetal undernutrition, which develops hypertension and cardiac alterations in adult life. In rats, suckling is the only source of nutrition until day 16 of lactation, while from day 17 to weaning there is an increasing content of chow in the stomach of the pups [28]. Therefore, we analyzed growth patterns during both periods of lactation. We found a sexual dimorphism in the growth pattern. In males, accelerated weight gain was observed during the period of exclusive lactation, while in female offspring catch-up growth was observed in the last week of lactation. We also observed that control rats fed on MUN mothers had a slower weight gain both in males and females. To explain these data, we considered differences in milk composition. However, we did not find significant variations in protein or fat content although a tendency toward lower values was observed in MUN dams. It is possible that the milk from MUN dams had lower macronutrient levels at birth, as we have evidenced in plasma [29], but they may normalize along lactation, since the rat returns to an *ad libitum* diet. In other rodent models of programming, milk protein content was found to be unaffected if the dam was fed with low protein [30,31] or with high energy diets [32] during gestation or lactation. We also observed a slower length gain during the last week of the lactation period, during which the rats suckle and also eat chow. We do not think this is related to the lower mineral content of the milk, since minerals seem to be relatively stable [13] and it is possible that there was a prior bone deficiency induced by undernutrition. We also discarded a lower milk yield due to compromised mammary gland development, observed in a rat model of fetal programming induced by uteroplacental insufficiency [33], since it would not explain the quicker weight gain in M-on-M offspring. Growth acceleration in MUN males could be explained by hyperphagia which can be programmed [34], is associated with increased neonatal growth rates and visceral adiposity [35], and it has been described in offspring from protein-restricted dams [36]. We do not think dam behavioral alterations, which have been described in SHR and WKY cross-fostering [37], explain our data since all the MUN and Control dams accepted the pups from the other mother successfully. The main influence on growth rate was observed during the period of exclusive lactation; since no major alterations in macronutrients were observed, we suggest the role of milk bioactive components, such as hormones or growth factors. This hypothesis is supported by metabolomic analysis of milk from rats exposed to mild caloric restriction during lactation showing changes in several compounds related to metabolic pathways [38], and the evidence that leptin supplementation during the suckling period reverses some of the metabolic alterations induced by a moderate maternal calorie restriction during gestation [39]. It would be interesting to conduct an in-depth milk metabolomic study on the alterations induced by undernutrition in our rat model.

Rapid growth gain has been related to increased adiposity and obesity development, and therefore, we analyzed possible alterations in body fat. We found that, in parallel with accelerated weight gain, by weaning M-on-M males exhibited hypertrophied perivascular WAT adipocytes compared to controls, an alteration reversed by nursing MUN offspring on a control dam. Similar results were found in a model of obesity programming induced by monosodium glutamate, where cross-fostering also mitigates obesity development [22]. In addition to energy-storing WAT, adipose tissue also comprises thermogenic brown adipose tissue (BAT), and beige adipocytes, an inducible form of thermogenic adipocytes interspersed within WAT [40] recruited postnatally in a process called browning [41]. Browning or “beigeing” has recently gained attention, since beige adipose tissue has a larger energy expenditure capacity, and induced browning in newborn rats decreases adipogenesis in adult life, suggesting a possible way through which the neonatal period can influence obesity development [42]. Therefore, we analyzed the proportion of beige adipocytes within the perivascular WAT in our experimental model, using Mitotracker, a specific dye for mitochondria, since beige adipocytes possess abundant cristae-dense mitochondria [40,43]. MUN rats had a very small proportion of beige-type adipocytes, which may be related to mitochondrial alterations induced by undernutrition, since mitochondrion is a very sensitive organelle and programming has been demonstrated in response to several intrauterine stress factors [44]. A reduction in beige thermogenic tissue could disbalance energetics and facilitate lipid accumulation, leading to the observed increased WAT adipocyte size. It was interesting that beige-type adipocytes increased by fostering MUN rats on a control mother. It has been demonstrated that beige adipose tissue appears spontaneously in WAT during early postnatal development with a peak of expression observed at 21 days [45]. Therefore, the lactation period is an important window during which adipose tissue type may be modulated. It would be interesting to analyze in MUN rats the effect of the lactation environment on adipocyte progenitor markers, such as mesenchymal cell antigen 1 (MSCA1), which has been positively correlated to obesity, adipocyte hypertrophy, and inflammation in children [46]. It is interesting to note that, increased browning occurred in MUN rats exposed to a control lactation environment, while it decreased in control rats exposed to a MUN mother. Therefore, unlike cardiovascular hypertrophy, which seems to require the first hit being during intrauterine life, adipose tissue may be regulated by the lactation environment alone. This is evidenced by our previous study showing that accelerated growth during lactation induced by reducing litter size, without prior programming, also increases WAT deposits [20]. Prolonged retention of thermogenic beige adipocytes maintains high whole-body energy expenditure and protects mice from diet-induced obesity [47]. Therefore, it would be interesting to evaluate if cross-fostered MUN rats retain this characteristic in adult life. Our finding of beige adipocytes within the WAT perivascular tissue is interesting, since, in mice, beige adipocytes are enriched within subcutaneous fat depots, and are rarely detected in visceral depots [43]. Perivascular WAT also plays a role in vascular tone and maintenance of normal structure [43], which is dysregulated by excess fat, as we have reported in obese mice [48]. A reduced proportion of beige-adipocytes may play a role in vascular dysfunction, an alteration observed in several animal models of fetal programming [49]. These alterations may be due to the pro-oxidant and proinflammatory environment associated with excess fat, favoring the local release of vasoactive factors. Thus, alterations in the proportion of beige-to-WAT perivascular adipose tissue may contribute to the association between obesity and cardiovascular disease [50], an aspect that deserves further attention. The cardioprotective effect of beige adipose tissue is suggested by the fact that individuals with detectable thermogenic adipose tissues have lower odds of hypertension and coronary heart disease [51].

Accelerated body growth gain in MUN males was paralleled by a hypertrophic response in the cardiovascular system observed in the aorta and the heart. The characteristics of the structural alteration in the aorta are compatible with an outward hypertrophic remodeling, with an increase in wall mass due to expansion of the external diameter. Females show a better adaptation, although the data also evidence a tendency towards a larger growth, as we have previously described in resistance arteries [52]. Undernutrition induced marked heart alterations by weaning, in addition to ventricular hypertrophy, a reduced diastolic function was also observed, suggesting that cardiac tissue has a high susceptibility to programming, also observed in guinea pigs exposed to undernutrition, which show a permanent deficiency in cardiomyocyte number [53]. We did not explore molecular markers of hypertrophy, such as BNP; we have evidence that this factor is not elevated in MUN offspring by weaning, but it is increased in aging in MUN males along with hypertension development and further cardiac hypertrophy [18]. The role of sex in heart alterations induced by undernutrition may be related to the lower efficiency of MUN male placenta, associated with poor vascularization [29], which may be particularly detrimental for the heart due to the unique feature of cardiomyocytes as non-dividing cells [54]. Cardiovascular hypertrophy in MUN males was rescued by cross-fostering, reinforcing the role of lactation in reprogramming cardiovascular disease. The importance of this period for future cardiometabolic health is supported by data in the genetic model of essential hypertension, the SHR rat, which reduces blood pressure levels in adult life if cross-fostered to a control dam [6]. It is worth mentioning that this model of essential hypertension also exhibits sexual dimorphism and we have found several common features with the MUN model [55].

## 5. Conclusions

The lactation period acts as a second hit consolidating programming initiated by fetal undernutrition in a sex-dependent manner, inducing cardiovascular and adipocyte hypertrophy.

The alterations programmed during fetal life may be counteracted during the perinatal period avoiding accelerated growth and thus, lactation and can be a window of intervention to reverse fetal programming.

The exclusive lactation period seems to play a key role in organ hypertrophy, despite no differences in breastmilk macronutrients, suggesting the role of bioactive factors, which deserves in-depth analysis.

## Figures and Tables

**Figure 1 biomedicines-10-02504-f001:**
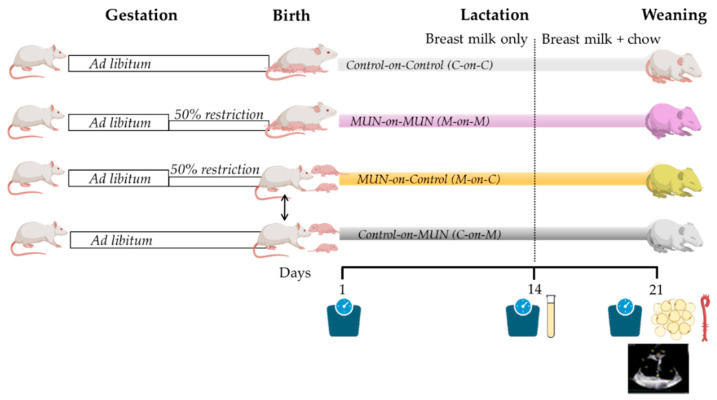
Experimental protocol diagram. MUN, maternal undernutrition during gestation; C-on-C, control pups fostered by a control mother (n = 6 dams); M-on-M, MUN rats fostered by their own mothers (n = 6 dams); C-on-M control rats fostered by a MUN mother (n = 4 dams); MUN-on-C, MUN rats fostered by a control mother (n = 4 dams).

**Figure 2 biomedicines-10-02504-f002:**
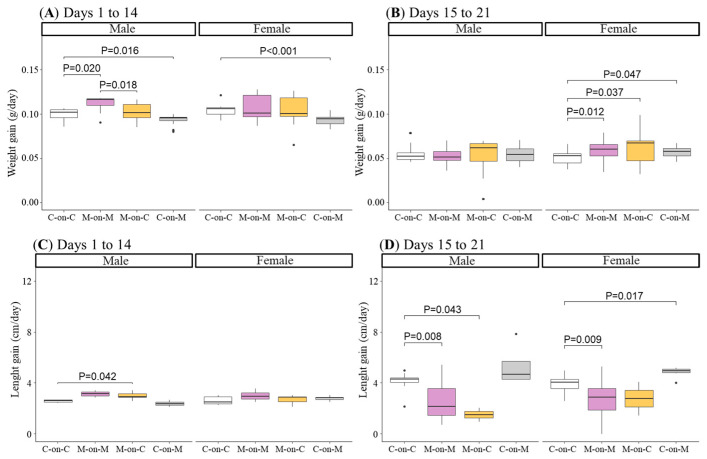
Body weight from day 1 to 14 of lactation (**A**) and from day 15 to 21 of lactation (**B**), and body length gain from day 1 to 14 of lactation (**C**) and from day 15 to 21 of lactation (**D**) in male and female rats from MUN and Control mothers. MUN, maternal undernutrition during gestation; C-on-C, control pups fostered by their mothers (34 males and 38 females); M-on-M, MUN rats fostered by their mothers (32 males and 40 females); C-on-M, control rats fostered by MUN mothers (21 males and 27 females); M-on-C, MUN rats fostered by a control mother (25 males and 23 females). Data show the median and interquartile range [Q1; Q3] and the showed *p*-Value (P) was extracted from Dunnett’s post-hoc pairwise comparison test when the Kruskal-Wallis test was P < 0.05. Dots show outliers.

**Figure 3 biomedicines-10-02504-f003:**
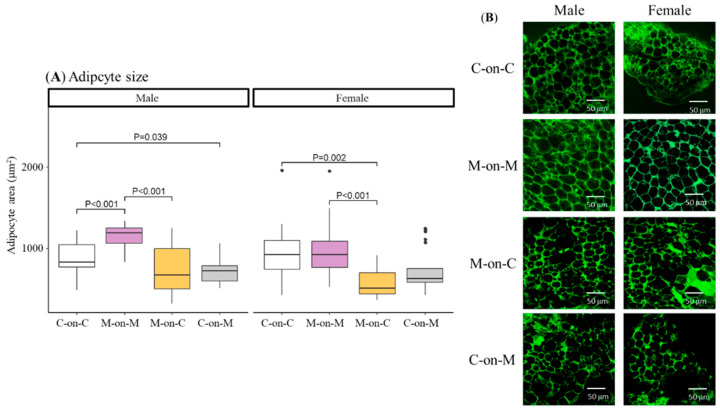
Average size of perivascular adipocytes (**A**) and representative images obtained by confocal microscopy (**B**) of 21-day-old male and female offspring from MUN and Control mothers. MUN, maternal undernutrition during gestation; C-on-C, control pups fostered by their mothers (6 males and 6 females); M-on-M, MUN rats fostered by their mothers (6 males and 6 females); C-on-M, control rats fostered by MUN mothers (6 males and 6 females); M-on-C, MUN rats fostered by a control mother (6 males and 6 females). Data show the median and interquartile range [Q1; Q3], and the showed *p*-Value (P) was extracted from Dunnett’s post-hoc pairwise comparison test when the Kruskal-Wallis test was P < 0.05. Images were acquired with a Leica TCS SP2 confocal microscope, ×40 objective at Ex = 488 nm/Em = 500–560 nm wavelength; scale bar = 50 μm. Dots shown outliers.

**Figure 4 biomedicines-10-02504-f004:**
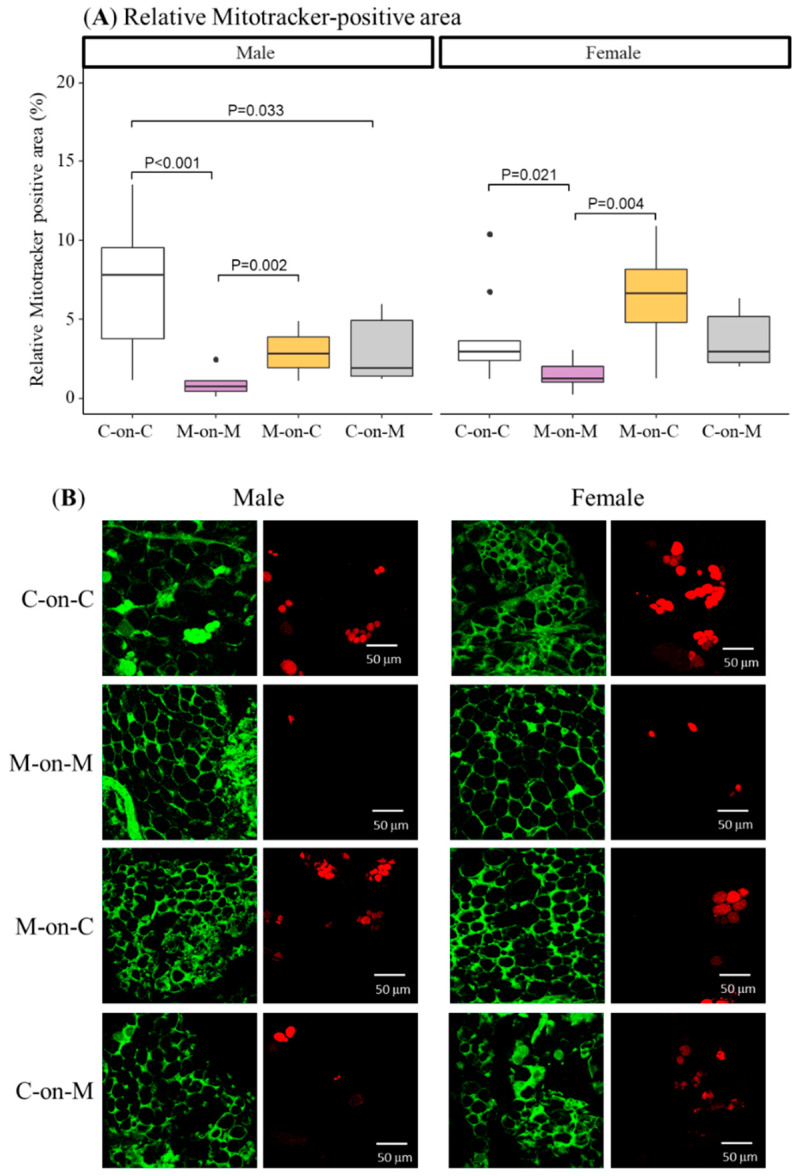
Percentage of Mitotracker-positive adipocytes in perivascular fat (**A**) and representative images obtained by confocal microscopy (**B**) of 21-day-old male and female offspring from MUN and Control mothers. MUN, maternal undernutrition during gestation; C-on-C, control pups fostered by their mothers (6 males and 6 females); M-on-M, MUN rats fostered by their mothers (6 males and 6 females); C-on-M, control rats fostered by MUN mothers (6 males and 6 females); M-on-C, MUN rats fostered by a control mother (6 males and 6 females). Data show the median and interquartile range [Q1; Q3], and the showed *p*-Value (P) was extracted from Dunnett’s post-hoc pairwise comparison test when the Kruskal-Wallis test was P < 0.05. Images were acquired with a Leica TCS SP2 confocal microscope, ×40 objective at Ex = 488 nm/Em = 500–560 nm (green images) and Ex = 581 nm/Em = 644 nm (red images); scale bar = 50 μm. Dots show outliers.

**Figure 5 biomedicines-10-02504-f005:**
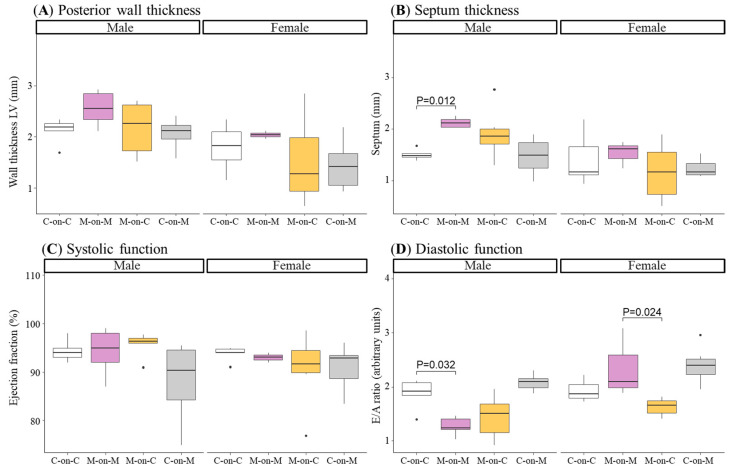
Posterior wall thickness (**A**), septum thickness (**B**), ejection fraction (**C**), and E/A ratio (**D**) of 21-day-old male and female offspring from MUN and Control mothers. MUN, maternal undernutrition during gestation; C-on-C, control pups fostered by their mothers (6 males and 6 females); M-on-M, MUN rats fostered by their mothers (6 males and 6 females); C-on-M, control rats fostered by MUN mothers (6 males and 6 females); M-on-C, MUN rats fostered by a control mother (6 males and 6 females); LV, left ventricle. Data show the median and interquartile range [Q1; Q3], and the showed *p*-Value (P) was extracted from Dunnett’s post-hoc pairwise comparison test when the Kruskal-Wallis test was P < 0.05. Dots show outliers.

**Figure 6 biomedicines-10-02504-f006:**
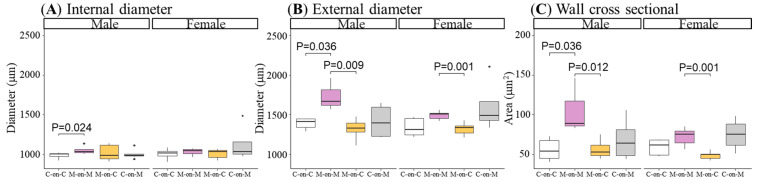
Internal (**A**) and external diameters (**B**), and wall cross-sectional (**C**) of the thoracic aorta in 21-day-old male and female offspring from MUN and Control mothers. MUN, maternal undernutrition during gestation; C-on-C, control pups fostered by their mothers (6 males and 6 females); M-on-M, MUN rats fostered by their mothers (6 males and 6 females); C-on-M, control rats fostered by MUN mothers (6 males and 6 females); M-on-C, MUN rats fostered by a control mother (6 males and 6 females). Data show the median and interquartile range [Q1; Q3], and the showed *p*-Value (*p*) was extracted from Dunnett’s post-hoc pairwise comparison test when the Kruskal-Wallis test was *p* < 0.05. Dot show outliers.

**Table 1 biomedicines-10-02504-t001:** End left ventricle dimensions of heart cycles of the offspring at day 21.

**Male**	**C-on-C**	**M-on-M**	**M-on-C**	**C-on-M**	** *p* ** **-Value**
End-diastolic diameter (mm)	2.92 [2.12; 3.07]	1.82 [1.75; 2.41]	2.55 [2.45; 2.55]	2.54 [2.49; 2.66]	0.358
End-systolic diameter (mm)	0.66 [0.49; 0.81]	0.36 [0.22; 0.36]	0.48 [0.44; 0.51]	0.84 [0.62; 1.06]	0.219
**Female**	**C-on-C**	**M-on-M**	**M-on-C**	**C-on-M**	** *p* ** **-Value**
End-diastolic diameter (mm)	2.51 [2.15; 3.10]	2.55 [2.48; 2.70]	2.52 [2.30; 3.01]	2.74 [2.13; 2.90]	0.536
End-systolic diameter (mm)	0.62 [0.48; 0.71]	0.66 [0.66; 0.66]	0.70 [0.60; 0.89]	0.73 [0.62; 0.78]	0.528

Data show median and interquartile range [Q1; Q3]. The *p*-Value was extracted from the Kruskal-Wallis test. Control (C); Maternal undernutrition (MUN).

**Table 2 biomedicines-10-02504-t002:** Summary of main findings.

		Male		Female	
		Alterations Induced by MUN	Reversal by CF in Lactation	Alterations Induced by MUN	Reversal by CF in Lactation
Body growth gain	Weight *from day 1 to 14*	Accelerated	Yes	ns	-
	Weight *from day 15 to 21*	ns	-	Accelerated	No
	Length *from day 1 to 14*	ns	-	ns	-
	Length *from day 15 to 21*	Decelerated	No	Decelerated	Yes
Adiposity	Size of adipocyte	Increased	Yes	ns	-
	% Mitotracker-positive	Decreased	Yes	Decreased	Yes
Heart structure and function	Posterior wall thickness	ns	-	ns	-
	Septum thickness	Increased	Yes	ns	-
	End-diastolic diameter	ns	-	ns	-
	End-systolic diameter	ns	-	ns	-
	Systolic function	ns	-	ns	-
	Diastolic function	Decreased	Yes	ns	-
Thoracic aorta structure	Internal diameter	Increased	Yes	ns	-
	External diameter	Increased	Yes	ns	-
	Wall cross-sectional	Increased	Yes	ns	-

Alterations induced by maternal undernutrition (MUN) are considered if M-on-M was different from C-on-C. Reversal by cross-fostering (CF) in lactation was considered if the alteration induced by MUN (M-on-C) is normalized, being similar to the C-on-C group. If no alterations were found, the effect of lactation was not considered (shown as -). No significant (ns).

## Data Availability

The data presented in this study are available on request from the corresponding author. The availability of the data is restricted to investigators based in academic institutions.

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
