# Peer review of "Slower Growth during Lactation Rescues Early Cardiovascular and Adipose Tissue Hypertrophy Induced by Fetal Undernutrition in Rats"

_biomedicines, 2022, doi:10.3390/biomedicines10102504_

Round 1

Reviewer 1 Report

This manuscript by Rodríguez-Rodríguez et al. aimed to investigate if slower postnatal growth during lactation improves early cardiometabolic alterations in offspring exposed to maternal undernutrition (MUN) during gestation. Pups were maintained with their own mother or cross-fostered during lactation. Body weight, length, cardiac size, aorta structure and adipocytes in white perivascular fat were measured at weaning. Major findings of the manuscript are that MUN males, but not females, showed reduced diastolic function and hypertrophied heart, aorta, and adipocytes, although beige-type adipocytes percentage was smaller in MUN males and females. Fostering MUN offspring on control dams significantly reduced weight gain rate, cardiovascular and fat hypertrophy, increasing beige-adipocyte proportion. Conversely, control offspring nursed by MUN mothers reduced body growth gain, without cardiovascular modifications. The authors interpreted these data to suggest that slower growth during lactation can rescue early cardiovascular alterations induced by fetal undernutrition. Overall, this is a reasonably well prepared manuscript. Experiments were thoroughly designed, statistical analyses are sound and data are clearly presented. Although the manuscript provides experimental evidence supporting the link between accelerated growth during early postnatal life in low body weight individuals and higher risk for cardiometabolic diseases, and that lactation period may be an effective window for intervention, they are descriptive in nature with limited mechanistic insights on the observations. Specific comments are provided below. 

Specific comments: 

1.      Abstract: Based on the experimental data the authors suggest lactation as intervention window, but they did not mention to whom the intervention is applied; mother or offspring (presumably to dams)? In addition, some elaboration on this narration in the text is suggested.

2.      Introduction: text in third paragraph on page 2 could be shortened, particular those on WAT and BAT. The arguments could be discussed in the Discussion section.

3.      Line 188, “mm)” should be (mm).

4.      Line 255, suggest to add a sentence to explain why measuring the mitochondria-positive cells.

5.      Line 235, why the data are expressed as the median and interquartile range [Q1; Q3], instead of the usual mean±standard deviation?

6.      Lines 254, 289, 293 and 455, change MUN-on-MUN males to M-on-M males.

7.      Lines 262, 263, 271 and 272 change MUN-on-C males to M-on-C males.

8.      Lines 309 and 330, “C-on C” males should be C-on-C males.

9.      Line 515, please add citations to the statement.

10.  Figures: asterisks in all figures are too small to be recognized. If they are not asterisks but individual datum point, why single them out?

11.  Figure 2A and D, why C-on-M pups have the lowest weight gain but highest length gain?

12.  M-on-M males have the largest adipocytes (cf. Figure 3) but the lowest beige-type adipocytes within perivascular WAT (cf. Figure 4). Why is this? Some explanation is required.

13.  Table 2: “” in the table means no change? If yes, please use no change.

14.  Discussion: Lactation environment seems to have differential impact on body, cardiovascular and adipose tissue growth in rats with LBW induced by undernutrition during fetal life. Some explanations and discussion on the discrepancies are recommended.

15.  Please revise the Discussion to shorten its length.

Author Response

This manuscript by Rodríguez-Rodríguez et al. aimed to investigate if slower postnatal growth during lactation improves early cardiometabolic alterations in offspring exposed to maternal undernutrition (MUN) during gestation. Pups were maintained with their own mother or cross-fostered during lactation. Body weight, length, cardiac size, aorta structure and adipocytes in white perivascular fat were measured at weaning. Major findings of the manuscript are that MUN males, but not females, showed reduced diastolic function and hypertrophied heart, aorta, and adipocytes, although beige-type adipocytes percentage was smaller in MUN males and females. Fostering MUN offspring on control dams significantly reduced weight gain rate, cardiovascular and fat hypertrophy, increasing beige-adipocyte proportion. Conversely, control offspring nursed by MUN mothers reduced body growth gain, without cardiovascular modifications. The authors interpreted these data to suggest that slower growth during lactation can rescue early cardiovascular alterations induced by fetal undernutrition. Overall, this is a reasonably well prepared manuscript. Experiments were thoroughly designed, statistical analyses are sound and data are clearly presented. Although the manuscript provides experimental evidence supporting the link between accelerated growth during early postnatal life in low body weight individuals and higher risk for cardiometabolic diseases, and that lactation period may be an effective window for intervention, they are descriptive in nature with limited mechanistic insights on the observations. Specific comments are provided below.

Specific comments:

  1. Abstract: Based on the experimental data the authors suggest lactation as intervention window, but they did not mention to whom the intervention is applied; mother or offspring (presumably to dams)? In addition, some elaboration on this narration in the text is suggested.

ANSWER: We would like to thank the reviewer for the suggestions and comments. Regarding your question, our data show that modification in lactation environment can rescue the phenotypic alterations induced by undernutrition during fetal life. These findings lead to the suggestion that maternal interventions in nutrition or lifestyle, modifying breastmilk bioactive components may have an impact on offspring growth. We have modified the last sentence in the abstract to clarify it and elaborated this aspect in the discussion (lines 38-39).

  1. Introduction: text in third paragraph on page 2 could be shortened, particular those on WAT and BAT. The arguments could be discussed in the Discussion section.

ANSWER: We agree and have eliminated this paragraph from the introduction and included it in the discussion.

  1. Line 188, “mm)” should be (mm).

ANSWER: Thank you for the revision. We have modified the text and corrected the typos.

  1. Line 255, suggest to add a sentence to explain why measuring the mitochondria-positive cells.

ANSWER: The reason is that beige adipocytes have large number of mitochondria, unlike WAT adipocytes and a mitochondrial marker (mitotracker) can be used to detect them. A sentence has been added for clarification (line 208-209).

  1. Line 235, why the data are expressed as the median and interquartile range [Q1; Q3], instead of the usual mean±standard deviation?

ANSWER: Thank you for your appreciation. We decided to express the data as median and interquartile range since the variables did not have a known distribution. This made us use non-parametric techniques to be more conservative in the statistical tests applied.

  1. Lines 254, 289, 293 and 455, change MUN-on-MUN males to M-on-M males.
  2. Lines 262, 263, 271 and 272 change MUN-on-C males to M-on-C males.
  3. Lines 309 and 330, “C-on C” males should be C-on-C males.

ANSWER: Thank you for the thorough revision. We have now modified the text and corrected the typos, according to your suggestions (points 6, 7, 8).

  1. Line 515, please add citations to the statement.

ANSWER: citation number 53 applied to this sentence. We have modified the sentence in the discussion and included the reference in the appropriate place.

  1. Figures: asterisks in all figures are too small to be recognized. If they are not asterisks but individual datum point, why single them out?

ANSWER:  We did not put asterisks for statistical significance, instead the p-value (P) is shown in the figures. The points are data outliers. We think it is more accurate to include them, since we do not have a known distribution of the variables. This was also the reason for using the median and interquartile range, rather than mean and standard error. 

  1. Figure 2A and D, why C-on-M pups have the lowest weight gain but highest length gain?

ANSWER:  Figures 2A and 2D represent different periods of lactation; 2A was data regarding body weight gain during exclusive lactation period, while 2D was related to length during the last week of lactation, where animals suckle milk and also eat by themselves.

  1. M-on-M males have the largest adipocytes (cf. Figure 3) but the lowest beige-type adipocytes within perivascular WAT (cf. Figure 4). Why is this? Some explanation is required.

ANSWER:  We suggest that the reduced beige-type adipocytes may be related to mitochondrial programming, which has been demonstrated in response to several intrauterine stress factors in various cell types. A reduction in beige adipocyte tissue could disbalance thermogenic and facilitate lipid accumulation, leading to the observed increased WAT adipocyte size. This aspect has been included in the discussion (lines 457-459).

  1. Table 2: “-” in the table means no change? If yes, please use no change.

ANSWER:  We have modified the legend of table 2 and it is now clear.

  1. Discussion: Lactation environment seems to have differential impact on body, cardiovascular and adipose tissue growth in rats with LBW induced by undernutrition during fetal life. Some explanations and discussion on the discrepancies are recommended.

ANSWER:  Yes, it seems that cardiovascular organs are particularly sensitive to the impact of undernutrition during fetal life, being this effect sex-dependent. We have now added some explanation, regarding this aspect in the discussion (lines 494-496).

  1. Please revise the Discussion to shorten its length.

ANSWER:  We have shortened the discussion eliminating redundant parts and focused it on the main aspects, adding relevant aspects suggested by the reviewer.

Reviewer 2 Report

The study entitled " Cross-fostering during Lactation Rescues Early Cardiovascular and Adipose Tissue Hypertrophy induced by Fetal Undernutrition in Rats" presents a very unique report of cardiometabolic diseases countering response through Cross-fostering lactation. They developed the maternal under-nutritional and cross fostering rats models. Authors assessed breast milk molecular composition, performed  transthoracic ecocardiography, aortic structure analysis,  and assessment of adipocyte size and browning.

Major 

Cardiovascular hypertrophy molecular markers as ANF, BNP , vascular endothelial growth factor , NAD-dependent growth factors may need to assess to give mechanistic conclusion

further need to assess MSCA1 as adipocyte hypertrophy molecular marker

Minor observations:

citation of previous protocol of cross-fostering methodology

correction Spelling

Trans thoracic 

Author Response

The study entitled " Cross-fostering during Lactation Rescues Early Cardiovascular and Adipose Tissue Hypertrophy induced by Fetal Undernutrition in Rats" presents a very unique report of cardiometabolic diseases countering response through Cross-fostering lactation. They developed the maternal under-nutritional and cross fostering rats models. Authors assessed breast milk molecular composition, performed  transthoracic ecocardiography, aortic structure analysis,  and assessment of adipocyte size and browning.

Major:

Cardiovascular hypertrophy molecular markers as ANF, BNP, vascular endothelial growth factor, NAD-dependent growth factors may need to assess to give mechanistic conclusion

ANSWER: We would like to thank the reviewer for the suggestions. Regarding your question, you are right, these are important cardiovascular hypertrophy markers. We have previously assessed hypertrophy markers measuring BNP in plasma from MUN and control rats at different age points (doi: 10.1371/journal.pone.0171544.g004, reference 18) and found that by weaning the levels are normal, but later on they are increased in MUN rats. We suggest that hypertrophy at the end of lactation is at an early stage which later on progresses along with hypertension development. Regarding VGEF, we have not measured it in the offspring, but in the placenta, wefound a reduced expression in MUN male placenta, suggesting a poor vascular development (doi: 10.3390/ijms22010237). We have added these aspects to the discussion (lines 498-504).

further need to assess MSCA1 as adipocyte hypertrophy molecular marker

ANSWER: This is an interesting point, since MSCA1 expression has been found to associate with adipocyte hypertrophy in children and it is also related to mitochondrial respiration alterations. This definitely deserves further attention as potential marker (lines 462-466). 

Minor observations:

citation of previous protocol of cross-fostering methodology

ANSWER: Citation has been added to methods section

correction Spelling

Trans thoracic

ANSWER: we have tried to correct all the typos. Regarding trans thoracic, we found that most paper write it as transthoracic, and we have maintained this spelling.